# Family and Child Characteristics Associated with Foster Care Breakdown

**DOI:** 10.3390/bs9120160

**Published:** 2019-12-16

**Authors:** Liliya A. Aslamazova, Rifkat J. Muhamedrahimov, Elena A. Vershinina

**Affiliations:** 1Department of Pedagogical Psychology, Adyghe State University, 208, Pervomayskaya st., 385000 Maykop, Russia; 2Department of Psychology, Saint-Petersburg State University, 7-9, Universitetskaya Emb., 199034 St. Petersburg, Russia; rjm@list.ru; 3Pavlov Institute of Physiology Russian Academy of Sciences, 6, Makarov’s Emb., 199034 St. Petersburg, Russia; ver_elen@mail.ru

**Keywords:** foster family, children, traumatic experience, intervention program, foster care breakdown

## Abstract

Studies examining the experience of children returned from foster care can reveal its causes and the severity of the psychological consequences, as well as the positive effect of psychological support on family functioning. Our research was aimed at the features of children and characteristics of foster families who refuse to continue parenting foster children. The study participants were comprised of families raising a foster child (Group One—182 families), and families who refused to continue parenting and returned the child (Group Two—19 families). The study was conducted using the “standardized interview for parents” and the “list of traumatic experiences of the child.” The results show that the strongest contributor to foster family breakdown was the degree of the child’s traumatic experience before placement (for Group One, 3.9 (1.15); Group Two, 6.1 (1.31), U = 395.0, *p* < 0.001) and the minimal participation of the family in an intervention program (the total number of program activities the family did not participate in; for Group One, 48.5 (28.27)%, Group Two, 95.5 (2.58)%, U = 67.5, *p* < 0.001). Our data expand ideas about the functioning of foster families who have taken children with significant traumatic experience and indicate the need to improve the quality of psychological and social support to foster families as an important factor in preventing secondary returns.

## 1. Introduction

In accordance with the National Strategy of Action for Children (1 June 2012, # 761), there has been intensive work on the foster family system in the Russian Federation. The number of children registered in the state bank for orphan children has been reduced from 71.2 thousand in 2016, to 49.9 thousand at the beginning of 2018 [1]. According to the State report on the situation of children and families with children in the Russian Federation, at the end of 2016, 87% of orphans had been placed in families. At the same time, there has been a growth in secondary returns: every year, about 1% of all transfers of children to a foster family are canceled (5198 children were returned in 2014; 5713 in 2015; and 5227 in 2016) [2]. Most returns occur on the initiative of the foster parents: in 2014, 4452 (86%) of the children were returned; in 2015, 4952 (87%); and in 2016, 4375 (80%). The most common reason given by foster parents for return are: unfavorable appearance; impaired development and behavior of the foster child (29% returns); multiple health problems (18%); poor heredity (10%); serious conflicts in the family connected with the reception of the child (10%); uncertainty about their own competence as a substitute parent (6%); and the negative impact of the foster child on biological children (5%) [3,4].

Studies on the effect of the permanence of foster placement on the well-being of foster children attests to the importance of stable placement for the child’s ability to form a healthy secure relationship with a caring adult [5,6,7]. Such a relationship has a long-term effect on the subsequent development and functioning of the child, including his/her ability to cope with stress, learn, and build relationships with other adults [8,9]. It has been shown that children who have experienced a break-up with a foster family experience deep stress, a sense of loss and non-belonging, and a fear of forming new relationships [10,11].

An analysis of the reasons for foster families’ refusal to continue identified both legal reasons (imperfection of legislation, lack of responsibility for refusing to bring up) [12], and reasons related to the peculiarities of the foster children (appearance, development, behavior, health, lack of social welfare skills, age-related difficulties) [13], or the substitute parents (low level of readiness to receive a child; inadequate motivation, especially financially oriented; mismatch of expectations and reality; disappointment, rejection of the child; confusion, emotional exhaustion, etc.) [12,13]. Studies revealed the severity of the psychological consequences of returns for foster children and substitute parents [13,14], and the positive effect of psychological support on the functioning of the substitute family [15,16,17]. At the same time, there is a lack of information on what conditions are conducive to stable and high-quality living conditions for foster children in Russian families, on the effectiveness of child and family support programs, and on the factors associated with foster care breakdown. The purpose of the current study was to explore the features of children and characteristics of substitute families in connection with the foster care breakdown experience.

## 2. Materials and Methods

### 2.1. Participants

This paper presents the results of a study of 201 foster families supervised by the Republic of Adygheya’s (Russian Federation) foster families’ assistance center. This group included families who continued to raise a foster child (Group One) and those who dropped out and returned the child (Group Two). In accordance with the legislation of the Russian Federation, before accepting the child into the family, all prospective parents take an 80.5 academic hour mandatory training program. The training program includes topics on legal and social aspects of fostering, medical care for children, child development, as well as the development and behavior of children with institutional experience. After completion of the program, parents are registered with the guardianship authority, the governmental placing agency that controls the foster family care system. All families receive professional support and intervention by the same foster families’ assistance center. No contacts of foster children with birth families were observed. The key characteristics of the foster families and the children are presented in Table 1.

The age of children (M [SD]) in Group One was 6.9 years (3.18) (range: 0.71–14.98); in Group Two, it was 9.3 years (2.99) (2.82–14.01). The age of admission of children to the institution was, correspondingly, 3.3 (2.36) (0.01–13.2) and 4.5 (1.85) (0.3–8.1); the length of stay in an institution was 3.6 (2.24) (0.01–11.0) and 4.8 (2.29) (1.1–11.4). The study was done two to four weeks after the child was admitted to the foster care family. The information on the foster care breakdown was obtained later based on the families’ assessment.

The data presented in our study were collected within routine regular work of the Center for Foster Families and were analyzed in order to understand the results of the professional activities provided in the Center. The work of the Center, including assessment of children and parents, is regimented by the Regulation on the Center for Foster Families. According to the Regulation, all foster parents signed a written form of the contract, which regulates services provided by the Center. All parents also sign the informed consent form, including consent for assessments of the foster child and family members (according to the Russian Federation Federal Law (“On Personal Data”, No. 152, 2006).

**Ethics approval and participation consent:** All procedures were carried out within the framework of support of foster families by the Center for Psychological, Medical, and Social Assistance (Maykop, Republic of Adygheya) in accordance with the legislation of the Russian Federation.

**Availability of Data and Materials:** The datasets used and analyzed during the current study are available from the corresponding author on reasonable request.

### 2.2. Research Tools

The 54 items of the “standardized interview for parents” [18] were used to examine the foster families’ socio-demographic characteristics, including the family’s place of residence and the availability of a separate room for the child. The “list of traumatic experiences of the child” [18], which consists of 20 events (for example, mother/father’s death, mother/father’s alcoholism, etc.), was applied to study what traumatic experience may have occurred in the child’s life before he/she was accepted into a foster family. The foster parents were interviewed about these traumatic events two to four weeks after the child was taken into the family.

### 2.3. Statistical Analysis

The frequency of occurrence of the foster families’ and children’s characteristics (in percent) between two groups were compared using the Chi-Square (x^2^) test. To compare the mean values of characteristics, the Mann–Whitney U test was used. The correlations between the characteristics were calculated using Spearman’s rank correlation coefficient. Binary logistic regression was employed to identify the family and child characteristics causing the foster care breakdown. All analyses were conducted using SPSS Inc. software.

## 3. Results

Groups One and Two differed in the distribution of four of the characteristics presented in Table 1:The financial situation of the substitute family (refusal is more frequent in families with a lower financial position; x^2^ (2, N = 201) = 15.59, *p* < 0.001);The number of traumatic situations suffered by the foster child (family breakdown is manifested in a larger number of them; x^2^ (7, N = 201) = 84.67, *p* < 0.001);The degree of family participation in the intervention program (the minimum degree in 26.9% of families in Group One and 100% participation in Group Two; x^2^ (2, N = 201) = 41.04, *p* < 0.001); andAs a trend, the foster family’s location (more often when living in the countryside; x^2^ (1, N = 201) = 2.28, *p* < 0.10) (see Table 1).

There were no differences between the frequencies of distribution between the two groups (*p* > 0.10) for the other indicators listed in Table 1.

The age of the children in Group Two was higher than in Group One: (6.9 (3.18) vs. 9.3 (2.99); U = 957.0, *p* < 0.001). The children in Group Two had a higher age of admission to the institution (3.3 (2.36) and 4.5 (1.85); U = 1072.0, *p* = 0.006), as well as a longer period of institutionalization (3.6 (2.24) and 4.8 (2.29); U = 1126.0, *p* = 0.012). Due to the high correlation between the age of children at the time of study and (1) the age of their admission to the institution (in Group One, r = 0.74, *p* < 0.001, in Group Two, r = 0.537, *p* = 0.018), (2) the length of institutionalization (respectively, r = 0.659, *p* < 0.001 and r = 0.734, *p* < 0.001), and (3) the age of admission to a foster family (for both groups r = 1.00, *p* < 0.001), the age of children was subjected to further analysis.

At the next stage, binary logistic regression was employed, with the foster care breakdown experience as the dependent variable and the following characteristics as independent variables: family’s financial situation (groups of families with medium and high financial status were combined due to the relatively low occurrence of high income in both groups); place of residence; degree of participation in the intervention program (a percentage of the total number of program activities the family did not participate in; for Group One, 48.5 (28.27)%, Group Two, 95.5 (2.58)%, U = 67.5, *p* < 0.001); the number of traumatic situations the child experienced (Group One, 3.9 (1.15); Group Two, 6.1 (1.31), U = 395.0, *p* < 0.001); and the child’s age.

The analysis was completed after two steps of logistic regression, showing a 96.5% correct classification with the identification of the determinant family and the child’s characteristics of the foster care breakdown (see Table 2). After the first step, the probability of a greater number of traumatic situations in children from Group Two was 4.1 times higher than from Group One; β = 1.41, SE = 0.27, *p* < 0.001). After the second step, the probability of a greater number of traumatic situations in children from Group Two was 2.73 times higher than from Group One; β = 1.0; SE = 0.37; *p* = 0.007), and the probability of a greater percentage of non-participation in program activities for families from Group Two was 1.78 times higher than from Group One; β = 0.59; SE = 0.19; *p* = 0.002).

## 4. Discussion

The present study deals with the impact of the foster families’ and children’s characteristics on the continuation or rejection of foster care. The results show that the most significant contributors to foster care breakdown are the number of traumatic situations the child experienced before admission to a foster family and the minimal family participation in an intervention program.

Highlighting the foster child’s traumatic experience as a significant contributor to the failure of the foster care placement supports previous studies about the vulnerability of children who are in foster care [15] because they are victims of abuse and neglect [19]. These children experience more somatic disorders, emotional, behavioral, and educational difficulties [19], which significantly complicate their relationships with foster parents [17], complicate family integration [18], and increase the likelihood of failure [20]. Negative experience is a factor that provokes adaptive difficulties and behavioral disorders [21], which eventually become reasons for the foster parents’ reluctance to continue hosting the child [22,23].

Our results also indicate that families who returned their foster children rarely attended intervention program activities. Their inefficiency in caring for foster children is reflected in their passivity about participating in the support program [18]. Thus, the program team was limited in its ability to detect problems [24] and to include them in foster care breakdown prevention programs.

The first step of analysis showed that the refusal to continue as foster parents occurs more often in families with lower financial status, as well as those in rural areas. On the one hand, these data confirm previous findings that a foster family’s material well-being is a factor of its viability [25] and are consistent with data showing high levels of returns of foster children from rural families [26]. On the other hand, these data were not confirmed by the results of the logistic regression, which may indicate their contribution to the child’s upbringing continuation indirectly through the indicators identified by the regression analysis. Since foster care is associated with high financial expenses, low material wealth is a risk factor and in combination with other factors, can lead to the family abandoning the child. According to the literature, returns of children from rural families may also be associated with the distance of families’ residences from the support services, which would make participation in support activities more difficult [18,27]. However, the rural foster families in our study lived within 1 to 1.5 h of the center, which is relatively comparable to the access time for urban families. The proximity of the studied families to the foster families’ assistance center reduces the degree of isolation of rural families and the lack of community-based stimulus for them and decreases the possibly of the breakdowns. The results of the binary logistic regression reveal that the place of foster family residence is not a significant contributor to foster care breakdown.

Our study shows that the age of children rejected by foster families was higher than the age of children who were accepted, and correlates with both the age of the child’s admission to an orphanage and the duration of stay in that institution, as well as the age of admission to foster families. The literature identifies the older age of the child at the time of placement as an absolute risk factor for failure [24,28,29,30,31]. At the same time, the results also indicate an indirect influence of the age factor on the family’s decision to refuse to continue the child’s upbringing. The number of traumatic situations the child experienced may be higher in foster children of older age. The adaptation process in foster families might be improved if prospective parents were selected and matched to children based on their skill set and preferences, including a preference for children of a specific age. Unfortunately, the newly created foster family system in the Russian Federation does not provide this type of matching.

A child’s gender, and whether he or she had typical development or disabilities, did not appear as a significant indicator of the success of placement. Although the insignificance of the gender factor is consistent with the results of other studies [26], the insignificance of the second indicator, on the one hand, contradicts the literature, showing that health-related developmental disorders are a risk factor for raising and keeping a child in a family, and create a high need of the family for early intervention [16]; on the other hand, they are consistent with the general results in this area [24]. The ambiguity could be caused by differences in the features of health and development of the foster children studied by different authors, as well as by concomitant factors, such as when the health problems are accompanied by emotional and behavioral problems [28].

Finally, our results show that indicators such as family size, availability of a separate room for a foster child, and the educational level and employment of the foster parents are not significant determinants of the success or failure of foster placement. This is consistent with other studies on family composition’s effect on the success of foster care [32], as well as those on the relationship between the education of the parents and the probability of the child being returned [28].

Our results on the contribution of family characteristics to keeping the foster child in the family emphasize the need for the organizations responsible for the training and support of foster families to take them into account. We need organizational and legal frameworks for the comprehensive involvement of existing and new foster families in intervention programs, and new forms of support, including for families living in remote and rural areas. Training for foster parents should include not only information on the psychological trauma of orphans and children left without parental care but should also teach them how to deal with a traumatized child and establish the importance of participating in a professional support program. Moreover, the staff of the foster families’ assistance center should be trained to help children and their new parents cope with the children’s traumatic experiences, including a history of neglect and violence in the biological family and social and emotional deprivation in institutions. The newly created system should include a requirement that prospective foster parents be selected based on the children’s needs.

The limitations of this study include the fact that its results cannot be directly applied to foster children who have experienced a secondary return. Information about the children’s traumatic experiences before their admission to the family was obtained from the statements of the foster parents and may differ from expert evaluations (support staff).

## 5. Conclusions

Raising a foster child is a complex and painstaking process, an important component of which is creating high-quality stable family conditions for the child. Our results show that among the characteristics of both the children and foster families, the strongest contributors to foster care failure were the child’s traumatic experiences before foster care and the foster family’s minimum degree of participation in programs of psychological support. It seems promising to continue studying factors that increase the chances of foster care success, including foster family dynamics and the children’s attachment to their foster parents. The results can expand our scientific understanding of the functioning of foster families and provide guidance in improving the current foster care system in the Russian Federation.

## Figures and Tables

**Table 1 behavsci-09-00160-t001:** The number of research participants (N), including in Group One (n_1_, %) and Group Two (n_2_, %), according to their characteristics, and results of intergroup comparison (x^2^, р).

Sample Characteristics	N	Group 1		Group 2		x^2^
		n_1_	(%)	n_2_	(%)	
**Foster Families**						
Total	201	182	90.5	19	9.5	0.75
couples’ families	163	149	81.9	14	73.7	
single-parent families	38	33	18.1	5	26.3	
Financial situation (per family member per month)						15.59 ***
high (over 550 dollars)	12	11	6	1	5.3	
average (150–450 dollars)	157	148	81.3	9	47.4	
low (less than 150 dollars)	32	23	12.6	9	47.4	
Place of living						2.28 ^+^
city	36	35	19.2	1	5.3	
countryside	165	147	80.8	18	94.7	
Degree of participation in intervention program						41.04 ***
1-minimum	68	49	26.9	19	100	
2-average	63	63	34.6	0	0	
3-maximum	70	70	38.5	0	0	
Foster mother education	199	180	90.5	19	9.5	2.16
university	61	57	31.7	4	21.1	
incomplete university	5	5	2.8	0	0	
college	87	76	42.2	11	57.9	
school	46	42	23.3	4	21.1	
Foster father education	165	151	91.5	14	8.5	1.12
university	30	26	17.2	4	28.6	
college	110	102	67.5	8	57.1	
school	25	23	15.2	2	14.3	
Foster mother employment	199	180	90.5	19	9.5	1.38
works	115	106	58.9	9	47.4	
does not work	70	61	33.9	9	47.4	
retired	14	13	7.2	1	5.3	
Foster father employment	165	151	91.5	14	8.5	0.99
works	155	141	93.4	14	100	
does not work	2	2	1.3	0	0	
retired	8	8	5.3	0	0	
**Foster Children**						
Gender						0.05
Boys	100	91	50	9	47.4	
Girls	101	91	50	10	52.6	
Developmental status						1.14
typical	108	100	54.9	8	42.1	
with disability	93	82	45.1	11	57.9	
Separate room for child						0.15
yes	27	25	13.7	2	10.5	
no	174	157	86.3	17	89.5	
Number of traumatic situations						84.67 ***
1–4	138	136	74.7	2	10.5	
5–8	63	46	25.3	17	89.5	

^+^ = *p* < 0.10; *** = *p* < 0.001.

**Table 2 behavsci-09-00160-t002:** The results of binary logistic regression (β, SE, OR with 95% confidence limits, p).

Characteristics	Β	SE	OR (95% CI)
Step 1 Number of traumatic situations Constant	1.41 *** *−*9.21	0.27 1.53	4.10 (2.40*–*7.0)
Step 2 Number of traumatic situations Number of non-participation in the intervention program Constant	1.0 ** 0.59 ** *−*60.16	0.37 0.19 18.08	2.73 (1.31*–*5.68) 1.78 (1.24*–*2.61)

** = *p* < 0.01; *** = *p* < 0.001.

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
