# Peer review of "Family and Child Characteristics Associated with Foster Care Breakdown"

_behavsci, 2019, doi:10.3390/bs9120160_

Round 1

Reviewer 1 Report

This article presents an original research, is well structured, using a clear style, and justify properly the methodological choices made. The criteria of collecting information about children´s traumatic experiences only from the foster families, however, without considering other sources, namely the social workers point of view, reduces in part the paper impact as the authors recongnised in the limitations of the study.

Aspects that can be improved concern the implications for practice, that could be improved with in-depth development, for instance with the presentation of concrets ways of intervention with foster families. Implications for practice should also be related with the studies included in the paper review.

Participation could be removed from keywords because it is implicit in the word intervention program and it is not an essencial concept ot this reflection.

In Table 1 the expression «intact families» might be substitute by «couples families». 

Finally, the mention to adopted children in limitation point it's inappropriate once the study is focused in foster care and the authors never assumed the intention to compare both care answers.

I suggest the publication of the article subject to those minor amendments.

Author Response

Response to Reviewer 1 Comments

Point 1: Aspects that can be improved concern the implications for practice, that could be improved with in-depth development, for instance with the presentation of concrete ways of intervention with foster families. Implications for practice should also be related with the studies included in the paper review.

Response 1: We thank the reviewer for pointing out the additional implications for practice. The sentence was added at the page 7 (line 219-222):

“Also the staff of the foster families’ assistance center should be trained to help children and their new parents cope with the children's traumatic experiences, including a history of neglect and violence in the biological family, and social and emotional deprivation in institutions.”.

Point 2: Participation could be removed from keywords because it is implicit in the word intervention program and it is not an essential concept of this reflection.

Response 2: These edits have been made. We appreciate the reviewer's attention to detail. participation” is removed from keywords (page 1, line 26)

Point 3: In Table 1 the expression «intact families» might be substitute by «couples families». 

Response 3: These edits have been made. The expression «intact families» was substituted by «couples’ families» (see Table 1). 

Point 4: Finally, the mention to adopted children in limitation point it's inappropriate once the study is focused in foster care and the authors never assumed the intention to compare both care answers.

Response 4: We thank the reviewer for pointing out this error. We have removed ”etended to adopted children, including those adopted in other countries, or” (page 7, line 209) and added “foster” (“foster children”, line 224).

Reviewer 2 Report

The topic for this study is important and worthy of investigation. I am not convinced however that the right methodology was used for this kind of study nor do I think that the findings/conclusions support socio-demographic reasons for placement breakdown. I applaud the attempt to use quantitative approaches to give validity however, I think this kind of study would have benefitted from a smaller sample group backed up with qualitative data in the form of participant interviews. This may have raised issues at a higher ethical level with different permissions required but I think it would have given more meaning to the current available data

I am not totally persuaded by the data and therefore the argument it tries to support. Where it is more than feasible that placements would breakdown in situations where a child was very traumatised, I think the reason may have less to do with the demographic/socio-economic situation of the carer and more to do with the social psychology of their support networks - personal and professional;  For example, The study does not say how the children were matched to prospective carers, their skill set or preference. Carers who felt more able to support teenagers who were then allocated very young children or vice versa, may not have felt equipped for the care giving demands. What was the role of the placing agency to supervise and support carers and what was the carer's status as carers? (was there any registration process, code of conduct, mandatory continuing professional development via attendance at training and participation in supervision, terms of licence etc?).

Carer isolation may have been compounded by rurality but the lack of community based stimulus triggering adverse reactions in the child may have equally been as positive as it may have been detrimental to the stability of a placement. These questions are worthy of consideration but left unanswered by the study as it draws its conclusions. Placement breakdown may also have been exacerbated by contact with birth families. All these questions are unanswered and therefore mean that the study's findings run the risk of being unsupported

I suggest shrinking the data set and digging more deeply into the findings to support the argument and consider the socio-psychological implications as well as/instead of socio-demographic. For example, surely where there is attachment and where carers (from any socio-economic/demographic group) have 'claimed' the child, placements are less likely to break down? If carers have been poorly selected in the first instance and then badly matched to the need of the child, would this not have a catastrophic effect on the outcome in terms of a trajectory of likely placement breakdown. These variables need addressing for the study to have greater validity

Author Response

Response to Reviewer 2 Comments

Point 1: The topic for this study is important and worthy of investigation. I am not convinced however that the right methodology was used for this kind of study nor do I think that the findings/conclusions support socio-demographic reasons for placement breakdown.

Response 1: We appreciate the reviewer bringing our attention to this issue, and have edited the manuscript for clarity and accuracy (see below).

Point 2: I applaud the attempt to use quantitative approaches to give validity however, I think this kind of study would have benefitted from a smaller sample group backed up with qualitative data in the form of participant interviews. This may have raised issues at a higher ethical level with different permissions required but I think it would have given more meaning to the current available data.

Response 2: We thank the reviewer for bringing attention to this point. Since the research project has concluded, we cannot interview participants from the research sample.  However, we thank the reviewer for the suggestion to add the interview and obtain the qualitative data, and will plan to include the method of interview in the future studies.

Point 3: I am not totally persuaded by the data and therefore the argument it tries to support. Where it is more than feasible that placements would breakdown in situations where a child was very traumatized, I think the reason may have less to do with the demographic/socio-economic situation of the carer and more to do with the social psychology of their support networks - personal and professional; 

Response 3: We appreciate the reviewer's feedback and agree that the quality of the families’ personal and professional support networks might influence the foster care breakdown. In our project we were studying general characteristics of families including financial situation, place of living, number of foster parents and parents’ education and employment (presented in the Table 1) which were summarized as socio-demographic characteristics. Agreeing with the reviewer that the family social psychology characteristics might be influential we have edited our manuscript for accuracy and removed the notion “socio-demographic” from the text: page 1, line 16; page 2, line 62; page 6, line 212 (“family” was included). Also the sentence “The present study deals with the impact of the socio-demographic characteristics of foster families and foster children’s characteristics on the continuation or rejection of foster care” was changed to “The present study deals with the impact of the foster families’ and children’s characteristics on the continuation or rejection of foster care” (page 5, line 155).

Point 4: For example, the study does not say how the children were matched to prospective carers, their skill set or preference. Carers who felt more able to support teenagers who were then allocated very young children or vice versa, may not have felt equipped for the care giving demands.

Response 4: We thank the reviewer for pointing out this and agree that the best is for children to be matched to prospective foster parents. Unfortunately, this important requirement is not included in the newly created system of foster families in Russia. Accordingly, we studied a sample in which foster children and parents were not matched to each other. The following sentence was included into the discussion section of the manuscript: “The adaptation process in foster families might be improved if prospective parents were selected and matched to children, based on their skill set and preferences, including a preference for children of a specific age. Unfortunately, the newly created foster family system in the Russian Federation does not provide this type of matching” (page 6, lines 194-197).

Point 5: What was the role of the placing agency to supervise and support carers and what was the carer's status as carers? (was there any registration process, code of conduct, mandatory continuing professional development via attendance at training and participation in supervision, terms of licence etc?).

Response 5: We agree that it is important to describe the role of different agencies and the status of carers in the Russian Federation system of foster care. In the RF the center for support of foster families and the placing agency are different organizations with different tasks. After placing the child into the foster care family the placing agency controls the family, while the center provides professional support and intervention for children and families. In order to be a foster parent one must attend the special training course and pass post training testing. Unfortunately, there is no mandatory continuing professional development system and no professional supervision or licensing for foster parents in RF.

To clarify the training requirements for foster parents’ we included in the Participants section of the manuscript the following sentences (page 2, lines 69-75): “In accordance with the legislation of the Russian Federation before accepting the child into the family all prospective parents take an 80.5 academic hour mandatory training program. The training program includes topics on legal and social aspects of fostering, medical care for children, child development, as well as development and behavior of children with institutional experience. After completion of the program parents are registered with the guardianship authority, the governmental placing agency that controls the foster family care system. All families receive professional support and intervention by the same foster families’ assistance center.”

Point 6: Carer isolation may have been compounded by rurality but the lack of community based stimulus triggering adverse reactions in the child may have equally been as positive as it may have been detrimental to the stability of a placement. These questions are worthy of consideration but left unanswered by the study as it draws its conclusions.

Response 6: We thank the reviewer for bringing attention to this point. Since all families received professional support and intervention by the same foster families’ assistance center, the words “the small number of specialists in rural support services” were removed from the manuscript (page 6, line 179). The subsequent part of the Discussion section was changed (page 6, lines 181-186): “However, the rural foster families in our study lived within 1 to1.5 hours of the center, which is relatively comparable to the access time for urban families. The proximity of the studied families to the foster families’ assistance center reduces the degree of isolation of rural families and the lack of community-based stimulus for them, and decreases the possibly of the breakdowns. Results of the binary logistic regression reveal that the place of foster family residence is not a significant contributor to foster care breakdown”.

Point 7: Placement breakdown may also have been exacerbated by contact with birth families.

Response 7: This has been clarified. A new sentence was added to the Participants section (page 2, lines 75-76): “No contacts of foster children with birth families were observed”.

Point 8: All these questions are unanswered and therefore mean that the study's findings run the risk of being unsupported.

Response 8: We deeply appreciate the reviewer’s very constructive and helpful comments, and believe we have made the changes recommended by the reviewer.

Point 9: I suggest shrinking the data set and digging more deeply into the findings to support the argument and consider the socio-psychological implications as well as/instead of socio-demographic. For example, surely where there is attachment and where carers (from any socio-economic/demographic group) have 'claimed' the child, placements are less likely to break down?

Response 9: We agree that it is important to consider the socio-psychological characteristics of foster families and children, and made changes in the Conclusions section of the manuscript: “It seems promising to continue studying factors that increase the chances of foster care success, including foster family dynamics and the children’s attachment to their foster parents. The results can expand our scientific understanding of the functioning of foster families, and provide guidance in the current foster care system in the Russian Federation” (page 7, lines 233-237).

Point 10: If carers have been poorly selected in the first instance and then badly matched to the need of the child, would this not have a catastrophic effect on the outcome in terms of a trajectory of likely placement breakdown. These variables need addressing for the study to have greater validity

Response 10: We once again thank the reviewer for pointing out this and agree that the best is for prospective foster parents be selected and matched to children. Unfortunately, these requirements are not provided for the RF. Accordingly, we studied a sample in which foster parents were not selected based on the child’s characteristics and children and families were not matched to each other. The following sentence was included into the discussion section of the manuscript: “The adaptation process in foster families might be improved if prospective parents were selected and matched to children, based on their skill set and preferences, including a preference for children of a specific age. Unfortunately, the newly created foster family system in the Russian Federation does not provide this type of matching” (page 6, lines 193-197). Also the new sentence was added: “The newly created system should include a requirement that prospective foster parents be selected based on the children’s needs.” (page 7, lines 222-223).

Round 2

Reviewer 2 Report

I'd like to thank the authors for their consideration of my comments and for both their responses and their amendments. The whole piece has been significantly improved by explaining the current context for fostering in the RF - an important inclusion in an international journal. The structural factors clearly play a significant part I the efficacy and success of placements and this needed to be spelled out clearly to a reader without this systemic insight. Thank you for this. With this detail your data sits within context- which was missing in your first draft.

Author Response

Thank you for your comments.